# Novel diagnostic potential of miR-1 in patients with acute heart failure

Seyyed-Reza Sadat-Ebrahimi[1], Aysa Rezabakhsh[2,3], Naser Aslanabadi[1], Milad Asadi[4], Venus Zafari[5], Dariush Shanebandi[4], Habib Zarredar[5], Elgar Enamzadeh[1], Hamed Taghizadeh[1], Reza Badalzadeh[1,6]*

1 Cardiovascular Research Center, Tabriz University of Medical Sciences, Tabriz, Iran, 2 Hematology, Immune Cell Therapy, and Stem Cells Transplantation Research Center, Clinical Research Institute, Urmia University of Medical Sciences, Urmia, Iran, 3 Emergency Medicine & Trauma Care Research Center, Tabriz University of Medical Sciences, Tabriz, Iran, 4 Immunology Research Center, Tabriz University of Medical Sciences, Tabriz, Iran, 5 Tuberculosis and Lung Research Center, Tabriz University of Medical Sciences, Tabriz, Iran, 6 Physiology Research Center, Tabriz University of Medical Sciences, Tabriz, Iran

* rbadalzadeh2020@gmail.com

## Abstract

Data Availability Statement: All relevant data are within the paper and its Supporting Information files.

### Background

A number of circulating micro-ribonucleic acids (miRNAs) have been introduced as convincing predictive determinants in a variety of cardiovascular diseases. This study aimed to evaluate some miRNAs' diagnostic and prognostic value in patients with acute heart failure (AHF).

### Method

Forty-four AHF patients were randomly selected from a tertiary heart center, and 44 healthy participants were included in the control group. Plasma levels of assessed miRNAs, including miR -1, -21, -23, and -423-5-p were measured in both groups. The patients were followed for one year, and several clinical outcomes, including in-hospital mortality, one-year mortality, and the number of readmissions, were recorded.

### Results

An overall 88 plasma samples were evaluated. There was no significant difference in terms of demographic characteristics between the AHF and healthy groups. Our findings revealed that mean levels of miR-1, -21, -23, and -423-5-p in AHF patients were significantly higher than in the control group. Although all assessed miRNAs demonstrated high diagnostic potential, the highest sensitivity (77.2%) and specificity (97.7%) is related to miR-1 for the values above 1.22 (p = 0.001, AUC = 0.841; 95%CI, 0.751 to 946). Besides, the levels of miR-21 and -23 were significantly lower in patients with ischemia-induced HF. However, the follow-up data demonstrated no significant association between miRNAs and prognostic outcomes including in-hospital mortality, one-year mortality, and the number of readmissions.

**Funding:** This study was supported by the Tabriz University of Medical Sciences.

**Competing interests:** The authors have declared that no competing interests exist.

## Conclusion

The result of our study demonstrated that miR-1, -21, -23, and -423-5-p can be taken into account as diagnostic aids for AHF. Nevertheless, there was no evidence supporting the efficacy of these miRNAs as prognostic factors in our study.

## 1. Introduction

Acute heart failure (AHF) is one of the most life-threatening and burdensome cardiovascular diseases (CVD) in both developed and developing countries [1, 2]. The prevalence of heart failure in 2020 was over six million patients in the United States of America (U.S.A), which is estimated to rise to more than eight million patients by 2030 [3]. The timely and accurate diagnosis of AHF and distinguishing it from chronic heart failure (CHF) are of great importance [4–6]. However, it is not always feasible to make a definite diagnosis merely based on history, physical examination, and echocardiogram [7]. Therefore, further investigations to establish novel biomarkers are warranted in this regard.

Despite well-established biomarkers, including brain (B-type) natriuretic peptide (BNP) and N-terminal pro-BNP (NT-proBNP) for assigning the diagnosis of AHF, there are some limitations in this era to apply as reliable indicators. For instance, a series of patient characteristics and conditions such as older age, obesity, renal dysfunction, atrial fibrillation, thromboembolic events, etc., can affect serum levels of BNP/ NT-proBNP and may lead to false results [8, 9]. Moreover, a multicenter trial on 1100 high-risk subjects with systolic HF reported no beneficial effect of using NT-proBNP-guided strategy within the routine outpatient management of those patients [10].

Micro-RNAs (miRNAs) are small non-coding RNAs with 21-25-nucleotide to regulate various genes [11]. Since their discovery in 1993 in nematode Caenorhabditis elegans, their role in several physiological and pathological conditions has been widely studied [12–17]. After that, further studies introduce a spread of biological effects of miRNAs within the circulatory system, including their role in cardiac fibrosis and hypertrophy, which are essential HF pathophysiologies [18]. Some miRNAs are closely related to the pathophysiology of HF.

MiR-1 is one of the most abundant miRNAs in cardiac tissue and is linked to cardiac remodeling being inversely correlated with hypertrophy. Experimental animal studies showed that adenoviral delivery of miR-1 was able to reverse hypertrophy [19, 20]. It plays also a role in atrial and ventricular arrhythmias. For example, it has been shown that its levels are markedly reduced in patients with atrial fibrillation. It also plays a role in acute coronary syndromes, a hypothesis that has yet to be further validated [21].

MiR-21 plays a role in inflammation and smooth muscle proliferation and fibrosis. It has been found to be increased in mice hearts subjected to ischemia-reperfusion and more specifically in fibroblasts within the infarct zone [19, 21]. It plays also a profibrotic role in heart failure from causes other than ischemia (i.e. aortic stenosis) through induction of matrix metalloprotease-2 (MMP-2) production [19, 20]. On the other hand, it attenuates cardiomyocyte apoptosis secondary to oxidative stress [20].

MiR-23 is upregulated in cardiac hypertrophy and promotes hypertrophy, while antago-miR-mediated silencing of miR-23a reverses it [20, 21].

Finally, miR-423-5-p has been also found to be increased in HF. In a small study, Tijsen et al. showed that miR-423-5-p strongly discriminated HF-associated dyspnea from healthy controls (C-statistic 0.91) and non-HF dyspnea (C-statistic 0.83) and was correlated with NT-

proBNP and LVEF [22]. In another study comparing several miR levels in patients with dilated cardiomyopathy and age- and sex-matched healthy controls, miR-423-5-p was found to have modest discrimination power (C-statistic 0.67) and was also correlated with NT-proBNP levels [23]. A 2020 systematic review showed that miR-423-5-p was among 5 miRNAs that were differentially expressed in more than one included studies [24]. Furthermore, miR-423-5-p levels in plasma may be time-dependent. A study of 246 post-MI patients showed that miR-423-5-p levels rose over time from index hospitalization to 3 months and up to 1 year post-discharge [25].

On the role of miRNAs in HF prognosis, a very recent systematic review and meta-analysis showed that HF patients with low expression of miR-1, miR-21, miR-23, and miR-423-5-p have significantly worse overall survival. However, among them, miR-423-5-p is the stronger biomarker for HF prognosis [26].

Many experimental animal, preclinical, and clinical human studies and reviews have tried to explore the role of miR in HF diagnosis with much attention focused on their relative superiority or non-inferiority compared to established biomarkers such as BNP or NT-proBNP [20, 21]. MicroRNAs may predict the development of HF at an earlier stage as they play a pathophysiological role in myocardial hypertrophy, fibrosis and apoptosis prior to overt clinical symptoms. Despite the positive results of some studies, the clinical utility of these miRNAs in clinical practice is yet to be determined. Here, we aimed to evaluate the potential of selected miRNAs, including miR-1, -21, -23, and -423-5-p, in the diagnosis and prognosis of AHF patients. These miRNAs were mainly selected as the most representative miRNAs that play different roles in cardiovascular pathology: hypertrophy, inflammation, fibrosis, acute coronary syndrome, and arrhythmia.

## 2. Methods

This study consisted of two phases. In the first phase, the plasma samples of the AHF patients and the healthy control group were evaluated to detect the levels of assessed miRNAs, including miR -1, -21, -23, and -423-5-p. Also, baseline demographic and clinical characteristics associated with the level of miRNAs were recorded. In the second phase, patients were followed for one year after admission, and several outcomes, including in-hospital mortality, one-year mortality, number of readmissions, and Functional Classification of New York Heart Association (FC) after one year, were recorded (**S1 Fig**).

### 2.1. Inclusion/Exclusion criteria

Patients who were admitted to a tertiary hospital with a diagnosis of either acute decompensation of CHF (International Classification of Diseases version 10 [ICD-10] code, I50.23: acute on chronic systolic [congestive] heart failure) or *de novo* AHF according to Framingham-based criteria following the confirmation of two attending cardiologists were included. Patients with HF with reduced ejection fraction (HFrEF: at least one major Framingham-based criteria/two minor criteria and ejection fraction [EF] <40%) and those with HF with preserved ejection fraction (HFpEF: at least one major criteria/two minor criteria and EF ≥50%) were included. As per established HF guidelines, heart failure with mid-range EF (HFmrEF) was defined as HF with an EF of 40–49%, and those patients with HFmrEF were excluded from our study [27]. Patients with HFmrEF were excluded as this phenomenon is considered a gray zone that shares some characteristics with HFpEF patients and some other characteristics with HFrEF patients, and could influence the outcome. Finally, 44 patients were randomly selected from AHF patients, and 44 healthy control were selected from the hospital staff or patients' companions.

## 2.2. Plasma sample

Five-milliliter blood samples were obtained within 24 hours after admission by a direct venous puncture into sodium citrate-containing tubes. The samples were stored at -80°C and were subjected to defrost/unfreeze.

## 2.3. RNA isolation

The process of RNA extraction was conducted using Tripura isolation reagent (Roche Inc., Germany), following the manufacturer's protocol. Agarose gel electrophoresis was used to assess the integrity of the extracted RNA. Moreover, the extracted RNA was further evaluated by NanoDrop 2000c UV-Vis spectrophotometer (Thermo, MA, USA).

## 2.4. Deoxyribonucleic acid (DNA) syntheses and real-time polymerase chain reaction (RT-PCR)

Deoxyribonucleic acid (DNA) syntheses and real-time polymerase chain reaction (PCR) for quantifying the relative miRNA levels in blood samples in this study were conducted as previously described [28]. In brief, a miRCURY™ LNA™ miRNA RT Kit (Exiqon Inc, Massachusetts) was utilized for cDNA synthesis using 10 ng of the extracted RNA based on the manufacturer's instructions. The reactions were conducted at a final volume of 10 μl in a T100 Thermocycler system (Bio-Rad, USA). A light cycler 96 system (Roche Inc., Germany) was utilized to quantify the mature miRNAs and use the ExiLENT SYBR Green master mix (Exiqon Inc, Massachusetts) and miRNAs specific primer sets. PCR was repeated for the second time to control the procedure's integrity and normalized by U6 expression levels considering the required controls (such as non-template control and no reverse transcription). The thermocycling conditions for quantitative PCR were as follows:

1 cycle of 50°C for 2 min, 95°C for 10 min, 40 cycles of 30 sec at 94°C, and 30 sec at 60°C.

U6 small nuclear RNA was considered as the endogenous control for normalization. The $2^{-\Delta Ct}$ method (mentioned below) was used to determine the relative expression levels of target miRNAs.

ΔCt = *Ct target gene–Ct U6*

## 2.5. Ethical consideration

This study was conducted in accordance with the ethical guidelines of the Declaration of Helsinki, and the ethics committee approved the protocol. The ethical approval for this study was issued from the Tabriz University of Medical Sciences with the ethics committee number: IR. TBZMED.REC.1397.142. Written informed consent was obtained from all participants after a brief description of the aim of the study.

## 2.6. Statistical analysis

All categorical data were reported as frequency and percentage and numeric data as mean ± SD (**S2 Fig**). The level of selected miRNAs is presented as fold-change relative to controls (mean level of miRNAs in the control group assumed to be equal to 1). Student t-test was used to compare two groups in data with normal distribution and the Mann-Whitney test for non-normal data or small sample size. ANOVA or Kruskal-Wallis tests were used to compare more than two groups in data with normal and non-normal distributions or small sample sizes, respectively. Receiver operating characteristic (ROC) curves were generated to estimate the area under the curve, sensitivity, and specificity. For the cutoff point values with the highest Youden index, sensitivity, specificity, positive predictive value (PPV), negative predictive value

(NPV), and accuracy were calculated. The association between the miRNAs' level and patients' outcome during follow-up was evaluated using multivariable logistic regression analysis by adjusting to the age, sex, smoking, and FC at admission.

The sample size was calculated using the formula of Buderer et al. [29] with an expected sensitivity of 0.95, expected specificity of 0.95, prevalence of 0.20, precision of 0.15, confidence level of 95%, and expected drop out of 15% resulting in a final sample size (with 15% dropout) of 49 patients [30].

## 3. Results

A total of 88 plasma samples were evaluated, including the plasma samples from 44 AHF patients and 44 healthy participants (S1 Fig).

No significant difference was seen between the two groups in terms of the baseline characteristics, including age, gender, body mass index (BMI), smoking, and place of residence (p>0.05; Table 1). Ischemic heart disease, hypertension, and diabetes were present in 17 (38.6%), 12 (27.2%), and 8 (18.1%) AHF patients, respectively. The mean EF in all AHF patients was $21.75 \pm 10.3\%$. Forty patients (90.9%) had HFrEF, and the majority of patients had the FC of III or IV at presentation (34.1 and 43.2, respectively). Twenty-nine patients

**Table 1. Baseline characteristics of groups of acute heart failure patients and healthy control.**

| | | AHF group (n = 44) | Healthy control (n = 44) | P value |
|---|---|---|---|---|
| Age $_{mean \pm SD}$ | | 55 (20) | 48 (15) | 0.106 |
| Sex $_{n (\%)}$ | Male | 33 (75.0) | 34 (77.3) | 0.803 |
| | Female | 11 (25.0) | 10 (22.7) | |
| Place of residence $_{n (\%)}$ | Urban | 31 (70.5) | 35 (79.5) | 0.325 |
| | Rural | 13 (29.5) | 9 (20.5) | |
| Body mass index $_{mean \pm SD}$ | | 23.55 (6.14) | 26.1 (4.38) | 0.060 |
| Smoking $_{n (\%)}$ | | 12 (27.3) | 8 (18.6) | 0.337 |
| Medical History $_{n (\%)}$ | Hypertension | 17 (38.6) | - | |
| | Diabetes | 12 (27.2) | - | |
| | Ischemic heart disease | 8 (18.1) | - | |
| | COPD | 3 (6.8) | - | |
| | Congenital myopathy | 2 (4.5) | - | |
| | Acute rheumatic fever | 2 (4.5) | - | |
| | CRF | 2 (4.5) | - | |
| | CVA | 2 (4.5) | - | |
| NYHA functional class at admission $_{n (\%)}$ | I | 0 (0) | - | |
| | II | 10 (22.7) | - | |
| | III | 15 (34.1) | - | |
| | IV | 19 (43.2) | - | |
| Type of heart failure (based on EF) $_{n (\%)}$ | Reduced EF | 40 (90.9) | - | |
| | Preserved EF | 4 (9.1) | - | |
| Type of heart failure (based on history) $_{n (\%)}$ | Acute on chronic | 28 (63.6) | - | |
| | de novo | 16 (36.4) | - | |
| Cause of heart failure $_{n (\%)}$ | Ischemic | 29 (65.9) | - | |
| | Non-ischemic | 15 (34.1) | - | |

AHF, Acute heart failure; EF, Ejection fraction, COPD, Chronic obstructive pulmonary disease CRF, Chronic renal failure; CVA, cerebrovascular accident; NYHA, New York Heart Association

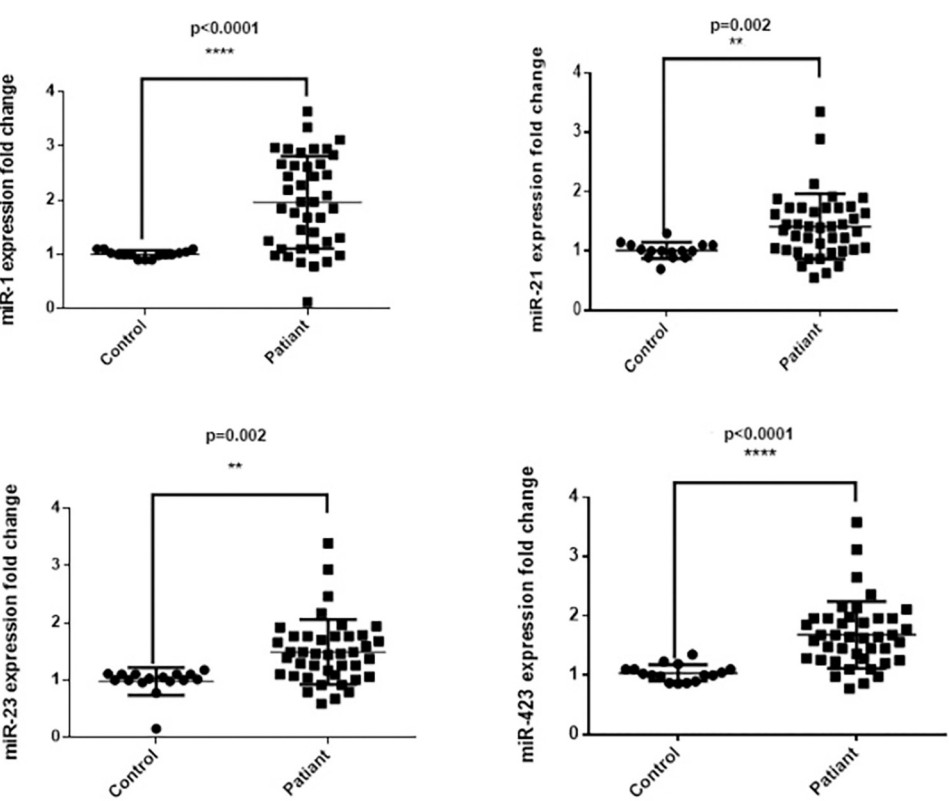

**Fig 1. The levels of four selected miRNAs in two groups (control and acute heart failure [AHF]).**

(65.9%) had AHF due to ischemia, and *de novo* AHF was detected in 16 patients (36.4%; **Table 1**).

The mean relative level of miR-1 in the AHF group was 1.930 ± 0.830 (p<0.00, **Fig 1**). Its sensitivity and specificity for the values above 1.22 were 77.2% and 97.7%, respectively (p = 0.001, AUC = 0.841; 95%CI, 0.751–946; **Table 2**; **Fig 2**). Relative miR-21 level in AHF patients was on average 1.445 ± 0.534 (P = 0.002) with the sensitivity and specificity for the values above 1.12 of 70.4% and 86.3%, respectively (p = 0.001, AUC = 0.857; 95%CI, 0.783–0.932). The mean relative level of miR-23 AHF was 1.490 ± 0.545 (p = 0.002). Its sensitivity

**Table 2. Overview of the diagnostic power of selected miRNAs for acute heart failure.**

| miRNA | Optimal cut-point value* | Group (n) | | AUC (95% CI) | Sensitivity (95% CI) | Specificity (95% CI) | PPV (95% CI) | NPV (95% CI) |
|---|---|---|---|---|---|---|---|---|
| | | AHF | Healthy control | | | | | |
| **miR-1** | ≥ 1.22 | 34 | 1 | 0.841 (0.75 to 94) | 77.2 (62.1 to 88.5) | 97.7 (87.9 to 99.9) | 97.14 (82.9 to 99.5) | 81.1 (71.3 to 88.1) |
| | < 1.22 | 10 | 43 | | | | | |
| **miR-21** | ≥1.12 | 31 | 6 | 0.857 (0.78 to 0.93) | 70.4 (54.8 to 83.2) | 86.3 (72.6 to 94.8) | 83.7 (70.5 to 91.7) | 74.5 (64.6 to 82.4) |
| | < 1.12 | 13 | 38 | | | | | |
| **miR-23** | ≥1.24 | 31 | 1 | 0.837 (0.74 to 0.92). | 70.4 (54.8 to 83.2) | 97.7 (87.9 to 99.9) | 96.8 (81.5 to 99.5) | 76.7 (67.6 to 83.9) |
| | < 1.24 | 13 | 43 | | | | | |
| **miR-423** | ≥ 1.35 | 32 | 3 | 0.881 (0.80 to 0.95) | 72.7 (57.2 to 85.0) | 93.1 (81.3 to 98.5) | 91.4 (77.9 to 96.9) | 77.3 (67.6 to 84.7) |
| | < 1.35 | 12 | 41 | | | | | |

* Relative miRNA level. miR, microRNA; AHF, Acute heart failure; PPV, Positive predictive value; NPV, Negative predictive value; AUC, Area under the curve

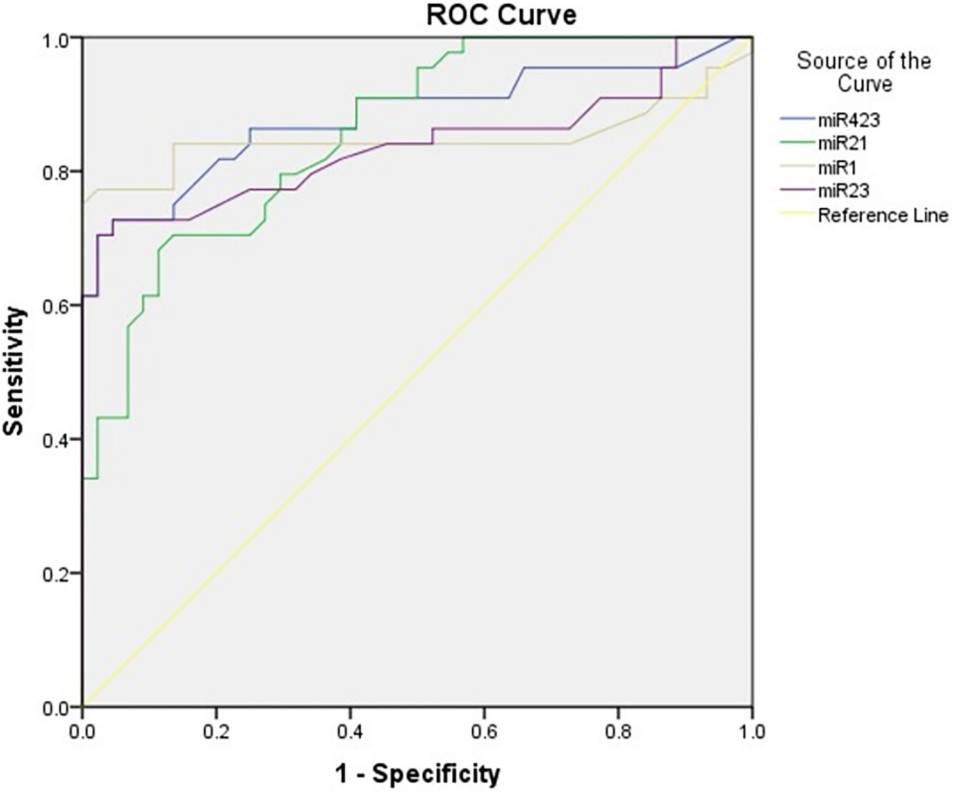

**Fig 2. ROC curve plotted for four selected miRNAs for diagnosis of acute heart failure.**

and specificity for the values above 1.24 were 70.4% and 97.7%, respectively (p = 0.001, AUC = 0.837; 95%CI, 0.746–0.928). Relative miR-423-5-p level in AHF patients was on average 1.681 ± 0.546 (P <0.001) with the sensitivity and specificity for the values above 1.39 of 72.7% and 93.1%, respectively (p = 0.001, AUC = 0.881; 95%CI, 0.805–0.957; **Table 2**; **Fig 2**). Accordingly, miR-1 had the highest net reclassification improvement index (NRI of miR-1, 0.35; 95%CI, 0.25 to 0.46; **S1 Table**).

Regarding the left ventricle function, the plasma levels of miR-21 and miR-423-5-p were higher in patients with HFpEF than in those with HFrEF, although these differences were not statistically significant (p = 0.775, p = 0.760, respectively). Also, the levels of miR-1 and miR-23 were lower in patients with HFpEF than those with HFrEF, which were not statistically significant (p = 0.353, p = 0.420, respectively) (**Fig 3A**). Also, no significant difference in the levels of studied miRNAs was detected between AHF patients with *de novo* and acute on CHF (**Fig 3B**). The miR-21 and miR-23 were significantly lower in patients with ischemia-induced HF when compared to the non-ischemic group (p = 0.027 and p = 0.029, respectively). However, no significant differences were observed in the levels of miR-1 and miR-423-5-p (**Fig 3C**).

Thirty-five patients were followed up (loss to follow-up = 20%). Of these patients, 3 patients (6.8%) died at their initial hospitalization, and 7 patients deceased during the follow-up (median, 1 month; ranging from 1 to 7 months after discharge). The leading cause of death was the decompensation of HF in all patients. Of the surviving patients (n = 25), 9 patients (25.7%) had not been hospitalized in one year after admission, and 16 patients (45.7%) had been readmission an average of 3 times. Four patients were candidates for heart

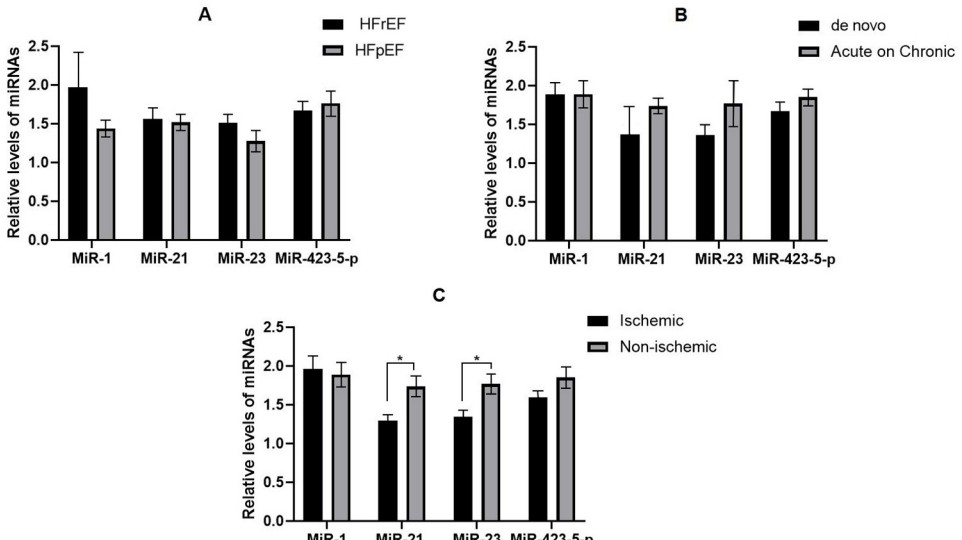

**Fig 3. The levels of selected miRNAs in different categories of acute heart failure.** A; The levels of selected miRNAs in two types of heart failure (heart failure with reduced ejection fraction [HFrEF] and with heart failure with preserved ejection fraction (HFpEF)), B; The levels of selected miRNAs with respect to the type of heart failure (de Novo vs acute on chronic), C; The levels of selected miRNAs with respect to the cause of heart failure (ischemia-induced vs non-ischemia induced).

transplantation, but no cardiac transplant was performed. Patients' NYHA FC after one year is reported in **S2 Table**. The majority of the surviving patients had an FC I (44%), indicating no physical activity limitations. There was no association between the level of miRNAs prognostic outcomes (**Table 3**).

## 4. Discussion

One of the main advantages of miRNAs, which are proposed as novel biomarkers for the diagnosis of a variety of CVD, is that they are detectable in various body fluids such as blood, saliva, urine, milk, amniotic fluid, etc. [31]. This feature enables us to conveniently detect the changes and track them down to find the etiologies without performing invasive procedures. Despite miRNAs possessing some advantages rather than the well-known biomarkers, e.g., NT-proBNP, they are not applied in the clinical setting. It is worth noting that some risk factors

**Table 3. The association between the level of miRNAs and the patients' outcomes in follow up.**

| miRNA | | In hospital mortality | Readmission in one year | Mortality in one year | FC after one year |
|---|---|---|---|---|---|
| **miR 1** | **OR (95%CI)** | 0.92 (0.11 to 7.69) | 0.99 (0.24 to 3.97) | 0.95 (0.36 to 2.54) | 0.07 (0.00 to 2.60) |
| | **P value** | 0.939 | 0.992 | 0.931 | 0.152 |
| **miR 21** | **OR (95%CI)** | 0.17 (0.00 to 8.62) | 1.25 (0.11 to 13.69) | 1.50 (0.24 to 9.24) | 0.06 (0.00 to 14.27) |
| | **P value** | 0.383 | 0.853 | 0.659 | 0.321 |
| **miR 23** | **OR (95%CI)** | 2.02 (0.06 to 68.04) | 1.03 (0.15 to 6.88) | 1.27 (0.27 to 5.97) | 0.33 (0.00 to 13.95) |
| | **P value** | 0.693 | 0.970 | 0.756 | 0.335 |
| **miR 423-5-p** | **OR (95%CI)** | 4.70 (0.14 to 150.74) | 0.02 (0.00 to 1.40) | 1.00 0.(95 to 1.04) | 0.10 (0.00 to 15.62) |
| | **P value** | 0.381 | 0.072 | 0.983 | 0.337 |

*Multivariate regression analysis by adjusting to age, sex, and FC at the first presentation. **Abbreviation:** FC, Functional Classification of New York Heart Association (NYHA); miR, microRNA; Events per variable ratios were 0.75, 3.2, and 2.5 for in-hospital mortality, readmission in one year, and mortality in one year, respectively.

indicated to affect the level of NT-proBNP, including age, obesity, gender, and renal function, have no substantial effect on the dynamic levels of prognostic/diagnostic miRNAs [32]. The current study investigated the clinical significance of the assessed miRNAs, including miR-1, -21, -23, and -423-5-p, in the diagnosis and prognosis of AHF. We found a significant elevation of these miRNAs in the plasma samples of AHF patients compared to the healthy control subjects, which display substantial diagnostic potential and significantly high sensitivity and specificity. The highest sensitivity and specificity were particularly observed in miR-1. Also, miR-23 had a similar specificity to miR-1 but the sensitivity was lower. MiRNAs may predict the development of HF at an earlier stage as they play a pathophysiological role in myocardial hypertrophy, fibrosis and apoptosis prior to overt clinical symptoms. In this way, the main application of miR in HF diagnosis may be in the form of a screening test. When designing a screening test, one needs a high sensitivity ($\geq$ 80%) in order to reduce the false negative rate. Otherwise, many diseased persons will be missed. Thus, the selection of the most appropriate cut-off criterion depends on the purpose of the diagnostic test. Modest sensitivity was detected for miR-1, miR-23, and miR-423-5-p. Nevertheless, all four miRNAs had remarkably high PPV, making them reliable for confirming AHF diagnosis. On the other hand, only the NPV of miR-1 was considerably high to rule out the patients with low probability of AHF with high accuracy.

A few studies, having investigated the diagnostic role of miR-1, -21, -23, and -423-5-p in AHF, yielded some controversial results [22, 33–36]. Unlike our findings, Corsten et al. failed to detect any dynamic changes in the level of miR-1 and -21 in AHF in comparison with healthy controls [32]. In parallel with this, Seronde et al. also postulated that the levels of miR-21 and -23 were similar among patients with AHF, stable CHF, and non-AHF with dyspnea [34]. Moreover, it has been documented that miR-1 was significantly lower among either AHF or stable CHF patients than non-AHF with dyspnea. Zhang et al. also demonstrated a significant diagnostic value of miR-21 for CHF with >99% sensitivity and 97.5% specificity [37]. Furthermore, a recent meta-analysis revealed that miR-423-5-p is an appropriate diagnostic biomarker for HF, with a pooled sensitivity of 81% and a pooled specificity of 67% [38]. Hereto, our findings further proved the diagnostic potential of these miRNAs in AHF patients.

Moreover, we explored the levels of miR-1, -21, -23, and -423-5-p in association with the ischemia-induced HF, left ventricle function, and the history of CHF (*de novo* or acute on chronic). However, we could detect no significant differences in the levels of assessed miRNAs, except for miR-21 and miR-23, which were significantly lower in patients with ischemia than in those with non-ischemic AHF. in line with our findings, Elis et al. also did not find any significant differences in the levels of miR-23 and miR-423-5-p between patients with HFrEF and HFpEF [33]. Likewise, Seronde et al. demonstrated no significant differences between *de novo* and acute on chronic AHF patients [34].

Notably, one-year follow-up of AHF patients did not show any efficient prognostic potential of the studied miRNAs regarding in-hospital mortality, one-year mortality, and the number of readmissions. Previously, Seronde et al. also failed to detect any prognostic value for miR-1, -21, -23, and -423-5-p in predicting readmission and one-year mortality [34]. Despite up-regulation of miR-21 during HF, Cakmak et al. also highlighted that it had no potential to be considered a prognostic value [39]; whereas Zhang et al. demonstrated a significant correlation of miR-21 with patients' two-year mortality but not with readmission rate [10]. Also, it has been reported that an increased level of miR-21 at the time of clinical compensation (chronic stable compensated state) was directly associated with better two-year survival and longer re-hospitalization-free status [40]. Noteworthy, higher miR-423-5p between the time of admission and clinical compensation was associated with fewer hospital readmissions in two years. Consequently, the time of acquisition of plasma samples can trigger a new development in the prognostic utility of miRNAs.

### 4.1. Study limitations

Some limitations are imposed on this study due to the challenges of detection and measurement of miRNAs. There are difficulties in measuring absolute levels of miRNAs (as opposed to relative expression vs controls), lack of standardized protocols, intra- and inter-laboratory variation, different types of samples (race, plasma, serum, and tissue), disease stage, etc. The exact time of sampling is crucial but it is yet to be standardized: i.e. miR-423-5-p may increase over time during follow-up. Also, the presence of arrhythmias (such as AF) may alter the levels of miR-1. Finally, in the case of acute coronary syndromes patients are usually on heparin; but heparin has been shown to be a significant inhibitor of PCR-based reactions (such as miRNA analysis) [41].

Since most of the patients had HFrEF in this study, the participation of the patients was completely random and not based on the HF type. In addition, the relatively small sample size in our study was another restriction in accurately evaluating the level of miRNAs among AHF patients with divergent characteristics. Also, we had a partly high rate of loss to follow-up (20%), reflecting presumable bias in the analysis of the patient's follow-up data. Indeed, 20 percent is suggested to be an acceptable rate for loss to follow-up [42]. On the other hand, limited financial support did not allow us to include more patients. Thus, events per variable ratios for some of our prognostic outcomes (particularly in-hospital mortality) were extremely low and the interpretations should be performed with caution and future studies with larger sample sizes are warranted to admit our findings for the prognostic value of these miRNAs.

Finally, it is suggested to compare AHF with other patients with dyspnea as well as healthy control groups in future studies.

## 5. Conclusion

In summary, our data demonstrated a remarkable diagnostic power of selected miRNAs (i.e., miR-1, -21, -23, 423-5-p) for AHF. Therefore, these miRNAs in particular miR-1 can be taken into account as diagnostic aids for AHF. Moreover, our results do not support the prognostic value of these miRNAs in AHF. Nevertheless, future large-scale studies are warranted to further elaborate on the prognostic value of these miRNAs. NT-proBNP is still the most convincing biomarker for AHF; therefore, our study might be considered a cornerstone for pioneering more investigations to establish the application of miRNAs as biomarkers for AHF detection in the clinic.

## Supporting information

**S1 Fig. Flowchart of inclusion of patients and follow up protocol.**
(TIF)

**S2 Fig. Flowchart of statistical analysis plan and approach.**
(TIF)

**S1 Table. The performance comparison and validation of models for each miRNA for diagnosis of acute heart failure.**
(DOCX)

**S2 Table. The level of miRNAs in patients who developed/not developed the adverse events during follow up.**
(DOCX)

**S1 Data.**
(XLSX)

## Author Contributions

**Conceptualization:** Seyyed-Reza Sadat-Ebrahimi, Naser Aslanabadi, Milad Asadi, Dariush Shanebandi, Habib Zarredar, Elgar Enamzadeh, Reza Badalzadeh.

**Data curation:** Aysa Rezabakhsh, Milad Asadi, Venus Zafari, Hamed Taghizadeh.

**Formal analysis:** Seyyed-Reza Sadat-Ebrahimi, Aysa Rezabakhsh, Dariush Shanebandi, Reza Badalzadeh.

**Funding acquisition:** Naser Aslanabadi, Reza Badalzadeh.

**Investigation:** Seyyed-Reza Sadat-Ebrahimi, Reza Badalzadeh.

**Methodology:** Seyyed-Reza Sadat-Ebrahimi, Naser Aslanabadi, Milad Asadi, Habib Zarredar, Hamed Taghizadeh.

**Project administration:** Seyyed-Reza Sadat-Ebrahimi, Venus Zafari.

**Resources:** Naser Aslanabadi, Dariush Shanebandi, Habib Zarredar, Hamed Taghizadeh, Reza Badalzadeh.

**Supervision:** Naser Aslanabadi, Dariush Shanebandi, Habib Zarredar, Elgar Enamzadeh.

**Validation:** Seyyed-Reza Sadat-Ebrahimi, Aysa Rezabakhsh, Naser Aslanabadi, Dariush Shanebandi, Habib Zarredar, Elgar Enamzadeh.

**Writing – original draft:** Seyyed-Reza Sadat-Ebrahimi, Aysa Rezabakhsh, Venus Zafari, Reza Badalzadeh.

**Writing – review & editing:** Seyyed-Reza Sadat-Ebrahimi, Aysa Rezabakhsh, Naser Aslanabadi, Venus Zafari, Dariush Shanebandi, Habib Zarredar, Elgar Enamzadeh, Reza Badalzadeh.

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
