## [Decision Letter · Decision Letter 0]

1 Jul 2022

PONE-D-22-15097Novel Diagnostic Potential of miR-1 in Patients with Acute Heart FailurePLOS ONE

Dear Dr. Badalzadeh,

Thank you for submitting your manuscript to PLOS ONE. After careful consideration, we feel that it has merit but does not fully meet PLOS ONE’s publication criteria as it currently stands. Therefore, we invite you to submit a revised version of the manuscript that addresses the points raised during the review process.

Specifically, check the statistics performed and adjust with proper methods. Report the event-per variable ration and the strategy to avoid overfitting. Please report the comparison for diagnostic accuracy in the ROC curves. 

We look forward to receiving your revised manuscript.

Kind regards,

Antonio Cannatà

Academic Editor

PLOS ONE

Journal Requirements:

a) Did participants provide their written or verbal informed consent to participate in this study?

5. Please ensure that you refer to Figure 1 in your text as, if accepted, production will need this reference to link the reader to the figure.

Additional Editor Comments:

Please clarify what the authors mean for ICD Implementation. Is it Implantation or upgrade? Also clarify the timing of ICD implantation and, when correcting, using it as a time dependent variable or removing it from the outcomes.

Please report the HFA-PEFF and the HF2PEF scores before classifying HFpEF.

Reviewers' comments:

Reviewer's Responses to Questions

**Comments to the Author**

1. Is the manuscript technically sound, and do the data support the conclusions?

Reviewer #1: Yes

Reviewer #2: No

2. Has the statistical analysis been performed appropriately and rigorously? 

Reviewer #1: I Don't Know

Reviewer #2: No

3. Have the authors made all data underlying the findings in their manuscript fully available?

Reviewer #1: Yes

Reviewer #2: No

4. Is the manuscript presented in an intelligible fashion and written in standard English?

Reviewer #1: No

Reviewer #2: Yes

5. Review Comments to the Author

Reviewer #1: May I congratulate the authors on conducting this original review. The reviewed manuscript fits with the Journal's scope, with a moderate clinical importance in the cardiology area and adds value to its field. In the review article entitled "Novel Diagnostic Potential of miR-1 in Patients with Acute Heart Failure", the authors try to assess the prognostic and diagnostic role of some miRNA in acute heart failure patients. The article summarises the risk factors and comorbidities of HFpEF and HFrEF patients and compares the plasma values for miR -1, -21, -23, and -423-5-p in acute heart failure patients and healthy subjects. The patients were followed for one year, including in-hospital mortality, one-year mortality, and the number of readmissions and the measures performed seem to have a diagnostic value.

The title is informative and relevant and the research/review question is clearly outlined, however, the authors should consider introducing HFmrEF in the title and in the text as well.

The background is clear and tries to shape what is already known about this topic, however it should offer a clearer view why did the authors only assessed this particular types of miRNA: -1, -21, -23, and -423-5-p for acute heart failure patients.

The main section needs an appropriate reporting: the data could be presented in a better way (data from the tables and figures could be more clearly presented).

Titles, columns and rows are labelled correctly and clearly, however, categories could be grouped more appropriately. The text is repetitive of the data included in tables, although the authors do not convince the reader what is a statistically significant result.

The conclusions section answers the aims of the study although it should be better supported by results. The manuscript should state its impact on current and future practice and also its limitations (eg: what is this assay influenced by, etc).

The references used are relevant, not so recent, referenced correctly and the authors tried to include appropriate key studies, although missed some.

Overall, the review was appropriate to answer the aim, based also on previous studies, however, the authors should address the following issues/ flaws of this article:

1. All acronyms and abbreviations must be explained in full at first mention (e.g DNA, ). All abbreviations used must be defined and spelt out in full in the caption.

2. Author(s) should be consistent when reporting abbreviation throughout the manuscript (eg: HfrEF (line 82) vs HFrEF (line 172,174); or HfpEF (line 83) vs HFpEF (line 172, 174, 186, 236, 254)

3. In the section "prevalence and incidence of HFpEF", in paragraph 3, starting with line 50, the reader could get confused as the authors start comparing HFpEF with HFrEF

4. In the section entitled "Risk factors and comorbidities of HFpEF", the authors should try and restate the ESC guidelines/ first sentence could be reworded.

5. Although the review should refer to HFpEF the authors compare HFpEF with HFrEF (line 50, 72, 96, 104, 113,135, 158, 165, 171, 230.232, 251,253,279,338, 339)

6. Authors should clarify why patients with HFmrEF= heart failure with mildly reduced ejection fraction where excluded from the study – line 84-85 (also that is not a sentence).

7. In study limitation, authors mention that most of the patients included in their study had HFpEF, however, as per the result section, most of them had HFrEF.

8. Authors should state the limitation more clearly (e.g.what influences the assay/ the test?)

Language:

1. "cronray" should read as "coronary" (line 63)?

2. “freeze-thawing” should read as “defrost/unfreeze”? (line 90)

3. A sentence should end with dot “.”(line 90)

4. A sentence always starts with capital letter. Authors should reconsider/ re-arrange first sentence in Discussions section. (line208). Also line 235.

5. Authors should reconsider second sentence in the conclusion section, line 266: “Therefore, they the miRNAs in particular miR-1 can be taken into account as diagnostic aids for AHF.”

Tables and figures:

1. For tables and figures, all the abbreviation used should be defined in a footnote (e.g. FC)

2. Any Table (Table 2) that require more than 2 pages, must be supplied as supplementary material.

3. When reporting in table 1 BMI that should be reported as 23.55 ±6.14

4. In the healthy control group all the subjects were non-smokers?

5. In table 1, when referred to Type of heart failure (based on history), n (%) - “acute on chronic” should read as” acute or chronic”?

6. Table 2 needs a better discussion in the text.

7. Authors should consider performing a statistical analysis plan and should consider using cluster approach.

Reference:

1. The first reference used is the guidelines, however it is advisable to use an updated version (current one dates from 2021)

2. The references used could be updated and more recent.

Reviewer #2: The authors submitted a research article in which they evaluated the potential of selected miRNAs, including miR-1, -21, -23, and -423-5-p, in the diagnosis and prognosis of AHF patients. They randomly selected 85 AHF patients and 58 healthy control individuals. The authors evaluated a signature of miRNAs (miR-1, miR-21, miR-423-5-p and miR-23) and did not find any association between the level of miRNAs prognostic outcomes, whereas diagnostic power of evaluated

miRNAs (i.e., miR-1, -21, -23, 423-5-p) for acute HF were determed. Although these findings are impressive, I would like to put forwrd several issues to comments.

1. Methodology: the authors should clearly report what HF was evaluated: chronic HF after recovery of acute HF or acute de novo HF / acutely decompensated HF. Endpoints are needed to be explained. Please, check and add more information. Flow chart with clear inclusion / non-inclusion criteria a re needed.

2. Statistics: sample size is required to be calculated. Please, add pertnent formula and an example of estimation. Yet, the validation method is needed to be reported along with comparissions of the models with AUCs, IDI and INR.

3. Results: Table 1 does not corresponds to the initial hypothesis of acute HF. The authors should re-check the data and improve them.

4. Please, use comparisson of the model with standard model that is required to be used in accordance to current ESC guideline

5. The multivariate log regression is needed to evaluate whether additional variables exist and influence prognosis.

6. PLOS authors have the option to publish the peer review history of their article (what does this mean?). If published, this will include your full peer review and any attached files.

Reviewer #1: No

Reviewer #2: No

---

## [Author Response · Author response to Decision Letter 0]

13 Aug 2022

Dr. Emily Chenette

Editor-in-Chief

PLOS ONE

Dear Dr. Chenette,

Subject: Revised Manuscript, Authors' Responses 

Thank you for your email and favorable response and comments. We appreciate the reviewers’ constructive suggestions. Please find attached the revised manuscript. 

Major revisions have been made to improve the manuscript, as suggested. 

The revisions and new citations have improved the content. Reformatting and editing also made a seamless flow of information.

The changes are marked by Track Changes. 

Below are the authors’ responses to the comments of the reviewers.

We hope that these revisions are acceptable and the manuscript can go forward for publication in PLOS ONE.

Thanks and Regards,

Reza Badalzadeh, PhD

Cardiovascular Research center

Madani Hospital, Tabriz University of Medical Sciences

Tabriz, Iran 

Tel: +984133373919

Fax: +984133373910

Email: reza.badalzadeh@gmail.com

Authors' Responses are colored in blue

Editor Comments:

Thank you for submitting your manuscript to PLOS ONE. After careful consideration, we feel that it has merit but does not fully meet PLOS ONE’s publication criteria as it currently stands. Therefore, we invite you to submit a revised version of the manuscript that addresses the points raised during the review process.

Specifically, check the statistics performed and adjust with proper methods. Report the event-per variable ration and the strategy to avoid overfitting. Please report the comparison for diagnostic accuracy in the ROC curves. 

The event-per-variable ratios for those with multivariable binary regression analysis were added to the footnote of Table 3. A statement regarding this issue was added to the limitation of the study section. Although it is believed that the evidence underlying the rule of EPV = 10 as a minimal sample size criterion for binary logistic regression analysis is weak [1].

Comparison for diagnostic accuracy in the ROC curves of each miRNA was added as a new column to Table 2.

Comment:

The manuscript format was changed to meet the style of PLOS ONE

Comment:

a) Did participants provide their written or verbal informed consent to participate in this study?

“Written informed consent was obtained from all participants.” This statement was added to the ethics statement section.

Comment:

Regarding the “Data Availability statement”, we added a reasonable phrase fit into this query, in the end of the manuscript before references, if it is not acceptable we can also prepare the minimal data set as supplementary files, if not, the required data could be uploaded in a standard repository system for public access. (https://data.4tu.nl/info/en/use/publish-cite/upload-your-data-in-our-data-repository) 

Comment:

The ORCID iD was added.

Comment:

5. Please ensure that you refer to Figure 1 in your text as, if accepted, production will need this reference to link the reader to the figure.

Figure 1 was referred to in the results section, paragraph 3.

Additional Editor Comments:

Please clarify what the authors mean for ICD Implementation. Is it Implantation or upgrade? Also clarify the timing of ICD implantation and, when correcting, using it as a time dependent variable or removing it from the outcomes.

The ICD implementation was removed from the outcomes.

Comment:

Please report the HFA-PEFF and the HF2PEF scores before classifying HFpEF.

Heart failure with preserved ejection fraction was defined based on left ventricular ejection fraction in this study. Considering that E/e′ ratio was not measured for our patients we are unable to report HFA-PEFF and the HF2PEF scores.

REVIEWER 1

Comment:

May I congratulate the authors on conducting this original review. The reviewed manuscript fits with the Journal's scope, with a moderate clinical importance in the cardiology area and adds value to its field. In the review article entitled "Novel Diagnostic Potential of miR-1 in Patients with Acute Heart Failure", the authors try to assess the prognostic and diagnostic role of some miRNA in acute heart failure patients. The article summarises the risk factors and comorbidities of HFpEF and HFrEF patients and compares the plasma values for miR -1, -21, -23, and -423-5-p in acute heart failure patients and healthy subjects. The patients were followed for one year, including in-hospital mortality, one-year mortality, and the number of readmissions and the measures performed seem to have a diagnostic value.

The title is informative and relevant and the research/review question is clearly outlined, however, the authors should consider introducing HFmrEF in the title and in the text as well.

We highly appreciate your comments and positive feedback. 

HFmrEF definition was added to the methods section and related reference was cited.

Comment:

The background is clear and tries to shape what is already known about this topic, however it should offer a clearer view why did the authors only assessed this particular types of miRNA: -1, -21, -23, and -423-5-p for acute heart failure patients.

A thorough explanation of the reasons for the selection of these miRNAs was added to the introduction section. These miRNAs were mainly selected to include the most representative miRNAs that play different roles in cardiovascular pathology: hypertrophy, inflammation, fibrosis, acute coronary syndrome, arrhythmia:

MiR-1 is one of the most abundant miRNAs in cardiac tissue and is linked to cardiac remodeling being inversely correlated with hypertrophy. Experimental animal studies showed that adenoviral delivery of miR-1 was able to reverse hypertrophy [2, 3]. It plays also a role in atrial and ventricular arrhythmias. For example, it has been shown that its levels are markedly reduced in patients with atrial fibrillation. It also plays a role in acute coronary syndromes, a hypothesis that has yet to be further validated [4].

MiR-21 plays a role in inflammation and smooth muscle proliferation and fibrosis. It has been found to be increased in mice hearts subjected to ischemia-reperfusion and more specifically in fibroblasts within the infarct zone [2, 4]. It plays also a profibrotic role in heart failure from causes other than ischemia (i.e. aortic stenosis) through induction of matrix metalloprotease-2 (MMP-2) production[2, 3]. On the other hand, it attenuates cardiomyocyte apoptosis secondary to oxidative stress [3].

MiR-23 is upregulated in cardiac hypertrophy and promotes hypertrophy, while antagomiR-mediated silencing of miR-23a reverses it [3, 4]. 

Finally, miR-423-5-p has been also found to be increased in HF. In a small study, Tijsen et al. showed that miR-423-5-p strongly discriminated HF-associated dyspnea from healthy controls (C-statistic 0.91) and non-HF dyspnea (C-statistic 0.83) and was correlated with NT-proBNP and LVEF [16]. In another study comparing several miR levels in patients with dilated cardiomyopathy and age- and sex-matched healthy controls, miR-423-5-p was found to have modest discrimination power (C-statistic 0.67) and was also correlated with NT-proBNP levels [5]. A 2020 systematic review showed that miR-423-5-p was among 5 miRNAs that were differentially expressed in more than one included studies [6]. Furthermore, miR-423-5-p levels in plasma may be time-dependent. A study of 246 post-MI patients showed that miR-423-5-p levels rose over time from index hospitalization to 3 months and up to 1 year post-discharge [7].

On the role of miRNAs in HF prognosis, a very recent systematic review and meta-analysis showed that HF patients with low expression of miR-1, miR-21, miR-23, and miR-423-5-p have significantly worse overall survival. However, among them, miR-423-5-p is the stronger biomarker for HF prognosis [8].

Many experimental animal, preclinical, and clinical human studies and reviews have tried to explore the role of miR in HF diagnosis with much attention focused on their relative superiority or non-inferiority compared to established biomarkers such as BNP or NT-proBNP [3, 4]. MicroRNAs may predict the development of HF at an earlier stage as they play a pathophysiological role in myocardial hypertrophy, fibrosis and apoptosis prior to overt clinical symptoms. Despite the positive results of some studies, the clinical utility of these miRNAs in clinical practice is yet to be determined. Here, we aimed to evaluate the potential of selected miRNAs, including miR-1, -21, -23, and -423-5-p, in the diagnosis and prognosis of AHF patients.

Comment:

The main section needs an appropriate reporting: the data could be presented in a better way (data from the tables and figures could be more clearly presented).

The design of the tables was improved to make them more clear. Nevertheless, upon you find it inadequate or have any specific suggestions please kindly guide us with more detailed explanation. 

Comment:

Titles, columns and rows are labelled correctly and clearly, however, categories could be grouped more appropriately. The text is repetitive of the data included in tables, although the authors do not convince the reader what is a statistically significant result.

The data in tables were grouped to show the results more clearly. The results section was improved accordingly.

Comment:

The conclusions section answers the aims of the study although it should be better supported by results. The manuscript should state its impact on current and future practice and also its limitations (eg: what is this assay influenced by, etc).

The conclusion was edited and the impact of the findings of our study was added. The limitations of measurement and detection of miRNAs as added to the study limitation section.

Comment:

The references used are relevant, not so recent, referenced correctly and the authors tried to include appropriate key studies, although missed some.

The references were updated accordingly.

Comment:

Overall, the review was appropriate to answer the aim, based also on previous studies, however, the authors should address the following issues/ flaws of this article:

1. All acronyms and abbreviations must be explained in full at first mention (e.g DNA, ). All abbreviations used must be defined and spelt out in full in the caption.

The abbreviations and acronyms were defined in the text and also the caption of Tabels.

Comment:

2. Author(s) should be consistent when reporting abbreviation throughout the manuscript (eg: HfrEF (line 82) vs HFrEF (line 172,174); or HfpEF (line 83) vs HFpEF (line 172, 174, 186, 236, 254)

The consistency of abbreviation was checked and the mentioned mistakes were corredcted..

Comment:

3. In the section "prevalence and incidence of HFpEF", in paragraph 3, starting with line 50, the reader could get confused as the authors start comparing HFpEF with HFrEF

We could not find this section in our manuscript. Would you please check again the section? It may have been already corrected or it could be related to another manuscript.

Comment:

4. In the section entitled "Risk factors and comorbidities of HFpEF", the authors should try and restate the ESC guidelines/ first sentence could be reworded.

We could not find this section in our manuscript. Would you please check again the section? It may be related to another manuscript.

Comment:

5. Although the review should refer to HFpEF the authors compare HFpEF with HFrEF (line 50, 72, 96, 104, 113,135, 158, 165, 171, 230.232, 251,253,279,338, 339)

We could not find the mentioned issue in the listed lines in our manuscript. Would you please check again the section?

Comment:

6. Authors should clarify why patients with HFmrEF= heart failure with mildly reduced ejection fraction where excluded from the study – line 84-85 (also that is not a sentence).

An explanation was added to the methods section.

Comment:

7. In study limitation, authors mention that most of the patients included in their study had HFpEF, however, as per the result section, most of them had HFrEF.

The sentence was edited.

Comment:

8. Authors should state the limitation more clearly (e.g.what influences the assay/ the test?)

The limitations section was completed.

Comment:

Language:

1. "cronray" should read as "coronary" (line 63)?

2. “freeze-thawing” should read as “defrost/unfreeze”? (line 90)

3. A sentence should end with dot “.”(line 90)

4. A sentence always starts with capital letter. Authors should reconsider/ re-arrange first sentence in Discussions section. (line208). Also line 235.

5. Authors should reconsider second sentence in the conclusion section, line 266: “Therefore, they the miRNAs in particular miR-1 can be taken into account as diagnostic aids for AHF.”

The language mistakes were corrected.

Comment:

Tables and figures:

1. For tables and figures, all the abbreviation used should be defined in a footnote (e.g. FC)

The abbreviations were accordingly completed.

Comment:

2. Any Table (Table 2) that require more than 2 pages, must be supplied as supplementary material.

Tables were edited to fit on one page. Thank you.

Comment:

3. When reporting in table 1 BMI that should be reported as 23.55 ±6.14

The style of reporting, “the mean ± SD” was corrected.

Comment:

3. In the healthy control group all the subjects were non-smokers?

Thank you for nice query. 

The prevalence of smoking among healthy subjects was added.

Comment:

4. In table 1, when referred to Type of heart failure (based on history), n (%) - “acute on chronic” should read as” acute or chronic”?

“Acute on chronic heart failure” was named according to the ICD-10-CM Code: I50. 23 as an acute heart failure occured following the chronic heart failure”.

Comment:

5. Table 2 needs a better discussion in the text.

The discussion over Table 2 was added to the discussion section. 

Comment:

6. Authors should consider performing a statistical analysis plan and should consider using cluster approach.

A flowchart of the statistical analysis plan and approach was added as supplementary figure 2.

Comment:

Reference:

1. The first reference used is the guidelines, however it is advisable to use an updated version (current one dates from 2021)

2. The references used could be updated and more recent.

The reference list was updated.

REVIEWER 2

The authors submitted a research article in which they evaluated the potential of selected miRNAs, including miR-1, -21, -23, and -423-5-p, in the diagnosis and prognosis of AHF patients. They randomly selected 85 AHF patients and 58 healthy control individuals. The authors evaluated a signature of miRNAs (miR-1, miR-21, miR-423-5-p and miR-23) and did not find any association between the level of miRNAs prognostic outcomes, whereas diagnostic power of evaluated miRNAs (i.e., miR-1, -21, -23, 423-5-p) for acute HF were determed. Although these findings are impressive, I would like to put forwrd several issues to comments.

Comment:

1. Methodology: the authors should clearly report what HF was evaluated: chronic HF after recovery of acute HF or acute de novo HF / acutely decompensated HF. Endpoints are needed to be explained. Please, check and add more information. Flow chart with clear inclusion / non-inclusion criteria are needed.

We included patients with acute decompensation of CHF (International Classification of Diseases version 10 [ICD-10] code, I50.23: Acute on chronic systolic (congestive) heart failure) or de novo acute heart failure. This explanation was added to the methods section.

Comment:

2. Statistics: sample size is required to be calculated. Please, add pertnent formula and an example of estimation. Yet, the validation method is needed to be reported along with comparissions of the models with AUCs, IDI and INR.

The sample size calculation method was added. Validation models including AUC and INR were also added to the supplementary table.

Comment:

3. Results: Table 1 does not corresponds to the initial hypothesis of acute HF. The authors should re-check the data and improve them.

The data in Table was re-checked and corrected.

Comment:

4. Please, use comparisson of the model with standard model that is required to be used in accordance to current ESC guideline

The AUC for each miRNA is reported for comparison with standard model.

Comment:

5. The multivariate log regression is needed to evaluate whether additional variables exist and influence prognosis.

The multivariate log regression has been already performed but due to the low number of outcomes and the proability of overfitting, we were not able to add more variables to our model. This issue was adderessed in our study limitations section.

References

1. van Smeden M, de Groot JAH, Moons KGM, Collins GS, Altman DG, Eijkemans MJC, et al. No rationale for 1 variable per 10 events criterion for binary logistic regression analysis. BMC Med Res Methodol. 2016;16(1):163. doi: 10.1186/s12874-016-0267-3.

2. Melman YF, Shah R, Das S. MicroRNAs in heart failure: is the picture becoming less miRky? Circ Heart Fail. 2014;7(1):203-14.

3. Wang H, Cai J. The role of microRNAs in heart failure. Biochimica et Biophysica Acta (BBA) - Molecular Basis of Disease. 2017;1863(8):2019-30. doi: https://doi.org/10.1016/j.bbadis.2016.11.034.

4. Romaine SPR, Tomaszewski M, Condorelli G, Samani NJ. MicroRNAs in cardiovascular disease: an introduction for clinicians. Heart. 2015;101(12):921-8.

5. Fan K-L, Zhang H-F, Shen J, Zhang Q, Li X-L. Circulating microRNAs levels in Chinese heart failure patients caused by dilated cardiomyopathy. Indian Heart J. 2013;65(1):12-6.

6. Peterlin A, Počivavšek K, Petrovič D, Peterlin B. The role of microRNAs in heart failure: a systematic review. Frontiers in Cardiovascular Medicine. 2020;7:161.

7. Bauters C, Kumarswamy R, Holzmann A, Bretthauer J, Anker SD, Pinet F, et al. Circulating miR-133a and miR-423-5p fail as biomarkers for left ventricular remodeling after myocardial infarction. Int J Cardiol. 2013;168(3):1837-40.

8. Yang J, Yang X-S, Fan S-W, Zhao X-Y, Li C, Zhao Z-Y, et al. Prognostic value of microRNAs in heart failure: A meta-analysis. Medicine. 2021;100(46).

---

## [Decision Letter · Decision Letter 1]

9 Sep 2022

Novel Diagnostic Potential of miR-1 in Patients with Acute Heart Failure

PONE-D-22-15097R1

Dear Dr. Badalzadeh,

We’re pleased to inform you that your manuscript has been judged scientifically suitable for publication and will be formally accepted for publication once it meets all outstanding technical requirements.

Kind regards,

Prof. Raffaele Serra, M.D., Ph.D

Academic Editor

PLOS ONE

Additional Editor Comments (optional):

amended manuscript is acceptable

Reviewers' comments:

Reviewer's Responses to Questions

**Comments to the Author**

1. If the authors have adequately addressed your comments raised in a previous round of review and you feel that this manuscript is now acceptable for publication, you may indicate that here to bypass the “Comments to the Author” section, enter your conflict of interest statement in the “Confidential to Editor” section, and submit your "Accept" recommendation.

Reviewer #1: All comments have been addressed

Reviewer #3: All comments have been addressed

2. Is the manuscript technically sound, and do the data support the conclusions?

Reviewer #1: Yes

Reviewer #3: Yes

3. Has the statistical analysis been performed appropriately and rigorously? 

Reviewer #1: I Don't Know

Reviewer #3: Yes

4. Have the authors made all data underlying the findings in their manuscript fully available?

Reviewer #1: Yes

Reviewer #3: Yes

5. Is the manuscript presented in an intelligible fashion and written in standard English?

Reviewer #1: Yes

Reviewer #3: Yes

6. Review Comments to the Author

Reviewer #1: All my questions and comments have been satisfactorily answered. I have no further questions and comments.

Reviewer #3: The authors provided a new version of their article. All comments have been addressed. Now the article is ready for publication.

7. PLOS authors have the option to publish the peer review history of their article (what does this mean?). If published, this will include your full peer review and any attached files.

Reviewer #1: No

Reviewer #3: No

---

## [Editor Report · Acceptance letter]

15 Sep 2022

PONE-D-22-15097R1 

Novel diagnostic potential of miR-1 in patients with acute heart failure 

Dear Dr. Badalzadeh:

I'm pleased to inform you that your manuscript has been deemed suitable for publication in PLOS ONE. Congratulations! Your manuscript is now with our production department. 

Kind regards, 

on behalf of

Prof. Raffaele Serra 

Academic Editor

PLOS ONE